# Improved Gradient Descent Optimization algorithm based on Inverse Model-Parameter Difference

## Abstract

A majority of deep learning models implement first-order optimization algorithms like the stochastic gradient descent (SGD) or its adaptive variants for training large neural networks. However, slow convergence due to complicated geometry of the loss function is one of the major challenges faced by the SGD. The currently popular optimization algorithms incorporate an accumulation of past gradients to improve the gradient descent convergence via either the accelerated gradient scheme (including Momentum, NAG, etc.) or the adaptive learning-rate scheme (including Adam, AdaGrad, etc.). Despite their general popularity, these algorithms often display suboptimal convergence owing to extreme scaling of the learning-rate due to the accumulation of past gradients. In this paper, a novel approach to gradient descent optimization is proposed which utilizes the difference in the model-parameter values from the preceding iterations to adjust the learning-rate of the algorithm. More specifically, the learning-rate for each model-parameter is adapted inversely proportional to the displacement of the model-parameter from the previous iterations. As the algorithm utilizes the displacement of model-parameters, poor convergence caused due to the accumulation of past gradients is avoided. A convergence analysis based on the regret bound approach is performed and the theoretical bounds for a stable convergence are determined. An Empirical analysis evaluates the proposed algorithm applied on the CIFAR 10/100 and the ImageNet datasets and compares it with the currently popular optimizers. The experimental results demonstrate that the proposed algorithm shows significant improvement over the popular optimization algorithms.

## 1 Introduction

Machine learning essentially involves implementing an optimization algorithm to train the model-parameters by minimizing an objective function, and has gained tremendous popularity in fields like computer vision, image processing, and many other areas of artificial intelligence Dong et al. (2016); Simonyan & Zisserman (2015); Dong et al. (2014).

Fundamentally, optimization algorithms are categorized into first-order Robbins & Monro (1951); Jain et al. (2018), higher-order Dennis & Moré (1977); Martens (2010) and derivative free algorithms Rios & Sahinidis (2013); Berahas et al. (2019) based on the use of the gradient of the objective function. First order algorithms use the first derivative to locate the optimum of the objective function through gradient descent. The second-order algorithms, on the other hand, use the second-order derivative information to approximate the gradient direction as well as the step-size to attain the optimum. Major disadvantage of these methods is the large computations required to determine the inverse of the inverse Hessian matrix. The quasi-newton algorithms like the L-BFGS solve this problem by approximating the Hessian matrix, and have gained significant popularity in many optimization problems Kochenderfer & Wheeler (2019).

Over the years, first-order optimization algorithms have become the primary choice for training deep neural network models. One of the widely popular first-order optimization algorithms is the stochastic gradient descent (SGD) which is (a) easy to implement and (b) is known to perform well across many applications Agarwal et al. (2012); Nemirovski et al. (2009). However, despite the ease

of implementation and generalization, the SGD often shows slow convergence due to the fact that SGD scales the gradient uniformly in all dimensions for updating the model-parameters. This is disadvantageous, particularly in case of loss functions with uneven geometry, i.e., having regions of steep and shallow slope in different dimensions simultaneously, and often leads to slow convergence Sutton (1986).

There have been a number of optimization algorithms developed over the years which attempt to improve the convergence speed of the gradient descent in general. There are algorithms that expedite the convergence in the direction of the gradient descent, which include the Momentum and the Nesterov's Accelerated Gradient (NAG) algorithms Qian (1999); Botev et al. (2017). Then there are algorithms that dynamically adapt the learning-rate of the algorithm based on an exponentially decaying average of the past gradients, and include the AdaGrad, RMSProp, and the Adadelta algorithms Duchi et al. (2011); Tieleman & Hinton (2012); Zeiler (2012). This category of algorithms implements the learning-rate as a vector, each element of whose corresponds to one model-parameter, and is dynamically adapted based on the gradient of the loss function relative to the corresponding model-parameter. Further, there are the adaptive gradient algorithms like the Adam algorithm and its variants like the Adamax, RAdam, Nadam, etc., which simultaneously incorporate the expedition of the gradient direction and the adaption of learning-rate based on the past gradients Kingma & Ba (2015); Dozat (2016); Reddi et al. (2018). Apart from that, some recent advances in the other first-order gradient descent methods include the signSGD Bernstein et al. (2018), variance reduction methods like the SAGA Roux et al. (2012), SVRG Johnson & Zhang (2013) and their improved variants Allen-Zhu & Hazan (2016); Reddi et al. (2016); Defazio et al. (2014).

The above-mentioned adaptive learning-rate methods, are amongst the most widely implemented optimizers for machine learning. However, despite increasing popularity and relevance, these methods have their limitations. Wilson et al. (2017) in their study showed that the non-uniform scaling of the past gradients, in some cases, leads to unstable and extreme values of the learning-rate, causing suboptimal convergence of the algorithms. A variety of algorithms have been developed over the last few years which improve the Adam-type algorithms further, like the AdaBound, which employ dynamic bounds on the learning-rate Luo et al. (2019); AdaBelief, which incorporates the variance of the gradient to adjust the learning-rate Zhuang et al. (2020); LookAhead, which considers the trajectories of the fast and the slow model-parameters Zhang et al. (2019); Radam, through rectifying the variance Liu et al. (2020); and AdamW, by decoupling the model-parameter decay Loshchilov & Hutter (2019).

In this paper, a new approach for the gradient descent optimization is proposed, where the learning-rate is dynamically adapted according to the change in the model-parameter values from the preceding iterations. The algorithm updates the learning-rate individually for each model-parameter, inversely proportional to the difference in model-parameter value from the past iterations. This speeds up the convergence of the algorithm, especially in case of loss function regions shaped like a ravine, as the model-parameter converging with small steps on a gentle slope is updated with a larger learning-rate according to the inverse proportionality, thereby speeding up the overall convergence towards the optimum. Further in the paper, a theoretical analysis to determine the lower and upper bounds on the adaptive learning-rate and a convergence analysis using the regret bound approach is performed. Additionally, an empirical analysis of the proposed algorithm is carried out through implementation of the proposed algorithm for training standard datasets like the CIFAR-10/100 and the ImageNet datasets on different CNN models.

The major contributions of the paper are as follows:

- A new approach to gradient descent optimization, which updates the learning-rate dynamically, inversely proportional to the difference in the model-parameter values from the preceding iterations is presented. The novelty of the algorithm lies in the fact that it does not employ an accumulation of past gradients and thus the suboptimal convergence, due to very large or very small scaling of the learning-rate, is avoided.

- A theoretical analysis determining the bounds on the adapting of the learning-rate is presented and a convergence analysis based on the regret bound approach is derived.

- An empirical analysis implementing the proposed algorithm on the benchmark classification tasks and its comparison with the currently popular optimization algorithms is performed.

The remainder of the paper is organized as follows: Section 2 gives a short background and formulates the problem at hand. In Section 3, the proposed algorithm is introduced and described in detail and the theoretical bounds on the adaptive learning-rate for a stable convergence are determined. In Section 4, the proposed algorithm is implemented on benchmark neural networks and its performance is evaluated in comparison with the currently popular optimization algorithms, like the SGD, Adam, AdaBelief, LookAhead, RAdam, AdamW, etc., which is followed by a conclusion and a brief outlook on the future aspects of the work in Section 5.

## 2 PROBLEM FORMULATION

### 2.1 NOTATIONS

Given a vector $p \in \mathbb{R}$, $p_k^{(i)}$ is used to denote the $i^{th}$ element of the vector at the $k^{th}$ time instance, $||p||_2$ denotes the $L2-$ norm of the vector and $|p|$ gives the absolute value of the vector. Given two vectors $p, q \in \mathbb{R}$, then $p \cdot q$ denotes the dot product, $p \odot q$ denotes the element-wise product and $p/q$ denotes the element-wise division of the two vectors.

### 2.2 BACKGROUND

Supervised learning involves training a neural network, initially modelled with parameters $\theta \in \mathbb{R}$, that maps the input dataset $x \in \mathbb{R}$ to predict an output $\hat{y} = f(\theta, x)$, and compute the gradient of a given loss function with respect to the model-parameters, and solve an optimization problem to adapt the parameters $\theta$ such that the loss function is minimized. The optimization (minimization) problem is represented as

$$\min_{\theta} \frac{1}{N} \sum_{i=1}^{N} J\left(y^{(i)}, f(\theta, x^{(i)})\right) \tag{1}$$

where $J$ is the loss function, $N$ is the number of training examples, $x^{(i)}$ is the $i^{th}$ training example, and $y^{(i)}$ is the corresponding labelled output. The SGD optimization recursively updates the model-parameters $\theta$ by subtracting the gradient of the loss function scaled by a certain factor (learning-rate) via the following update rule

$$\theta_{k+1} = \theta_k - \mu_k \cdot \hat{\nabla}_k(\theta_k) \tag{2}$$

where $\theta_k$ represents the model-parameters at instance $k$, $\hat{\nabla}_k$ is the unbiased estimate of the exact gradient of the loss function and $\mu_k$ is the learning-rate which is also generally adapted at every iteration Zinkevich (2003); Haykin (2014).

Slow convergence is one of the major challenges faced by the SGD optimization. SGD convergence can be particularly slow due to the complex geometry of the loss function. A variety of optimization algorithms, which dynamically adjust the learning-rate of the algorithm by considering an accumulation of the past gradients for adapting the learning-rate, have been developed and gained popularity in the past years Soydaner (2020); Dogo et al. (2018). The adaptive gradient algorithms, like Adam and its variants (as listed in section 1), despite their widespread popularity, display suboptimal convergence in many scenarios. This results from the non-uniform scaling of the learning-rate accumulation of the past gradients Wilson et al. (2017); Chen et al. (2019).Therefore, a novel approach to SGD optimization, which adjusts the learning-rate of the algorithm considering the difference in the model-parameter values instead of an accumulation of past gradients, is proposed in this paper.

## 3 PROPOSED ALGORITHM

The proposed algorithm is based on dynamic adjustment of the learning-rate according to the change in the values of the model-parameters between the preceding consecutive iterations. More precisely, the learning-rate for each model-parameter is dynamically adjusted by scaling it inversely proportional to the difference in the values of that model-parameter from the immediate preceding iterations. The proposed algorithm follows the norm that the model-parameters in regions of a gentle slope should be adapted with a larger learning-rate as compared to the parameters converging quickly over regions of steeper slope, thereby boosting the overall convergence of the algorithm towards the optimum.

The difference coefficient, denoted by $D_k^{(i)}$, is a function of the absolute difference between the values of the $i^{th}$ model-parameter from the immediate preceding iterations, and is given by

$$D_k^{(i)} = \frac{K}{1 + abs(\Delta\theta_k^{(i)})} \tag{3}$$

where $D_k^{(i)}$ is the difference coefficient of the $i^{th}$ model-parameter at the $k^{th}$ iteration and $\Delta\theta_k^{(i)} = \theta_k^{(i)} - \theta_{k-1}^{(i)}$ gives the difference between the values of the $i^{th}$ model-parameter between the $k^{th}$ and the $(k-1)^{th}$ iterations. The term $1 + abs(\Delta\theta_k^{(i)})$ limits the range of the denominator to $[1, \infty)$ so that extreme scaling of $\mu_k$ in case of very small $\Delta\theta_k$ is avoided. The learning-rate $\mu_k^{(i)}$ for the $i^{th}$ model-parameter is thus updated according to the following relation.

$$\mu_k^{(i)} = \mu_0 \cdot D_k^{(i)} = \mu_0 \cdot \left( \frac{K}{1 + abs(\Delta\theta_k^{(i)})} \right) \tag{4}$$

The characteristic of the $D_k^{(i)}$ determines the behavior of the dynamic learning-rate. Note that for $K = 1$, $D_k^{(i)} \in (0, 1], \forall \Delta\theta_k^{(i)} \in (-\infty, \infty)$. It can be observed that a large difference in the model-parameter values results in a smaller $D_k$ and thus a smaller adaptive learning-rate, while a smaller difference results in a larger $D_k$, i.e., a larger adaptive learning-rate.

The dynamic learning-rate, in this algorithm, is represented by a vector $M \in \mathbb{R}^L$, every element of whose gives the learning-rate corresponding to each model-parameter

$$M = \left[ \mu_k^{(1)}, \mu_k^{(2)}, ..., \mu_k^{(L)} \right]^T \tag{5}$$

where $\mu_k^{(i)} = \mu_0 \cdot D_k^{(i)}$ and $L$ is the number of model-parameters. Thus, at $k^{th}$ iteration, the update of the model-parameters is given by

$$\boldsymbol{\theta_{k+1}} = \boldsymbol{\theta_k} - \boldsymbol{M} \odot \hat{\boldsymbol{\nabla}}_k(\boldsymbol{\theta_k}) \tag{6}$$

The algorithm defined by the Eq. 6 above adjusts the learning-rate for each model-parameter as an inverse function of change in the model-parameter values. The Algorithm 1 depicts the generic framework of the proposed algorithm.

---

**Algorithm 1:** Proposed algorithm

---

**Input:** $x \in \mathbb{R}$
**Parameters:** initial model-parameters $\boldsymbol{\theta}_0$
                initial learning-rate $\mu_0$
**Iterate:**
 **for** k = 1,2,3,... until convergence
1.    compute gradient
     $\hat{\boldsymbol{\nabla}}_k = \partial\mathbf{J}(\boldsymbol{\theta}_k)/\partial\boldsymbol{\theta}_k$
2.    compute displacement coefficient
     $D_k^{(i)} = \frac{K}{\left(1 + abs(\Delta\theta_k^{(i)})\right)}$
3.    compute adaptive learning-rate
     $\mu_k^{(i)} = \mu_0 \cdot D_k^{(i)}$
     $M = \left[ \mu_k^{(1)}, \mu_k^{(2)}, ..., \mu_k^{(L)} \right]^T$
4.    update model-parameters
     $\boldsymbol{\theta}_{k+1} = \boldsymbol{\theta}_k - \boldsymbol{M} \odot \hat{\boldsymbol{\nabla}}_k(\boldsymbol{\theta}_k)$
 **end for**

---

The idea of utilizing the difference in the model-parameters is also used in the L-BFGS algorithm, which belongs to the category of quasi-Newton methods, and uses the history of past $m$ parameter

updates and their gradients, to estimate the Hessian of the objective function $H\left(\theta_k\right) = \nabla^2 J\left(\theta_k\right)$ Liu & Nocedal (1989); Kochenderfer & Wheeler (2019). In comparison, the proposed algorithm utilizes the difference in model-parameters to update the learning-rate of the gradient descent. Also, the BFGS algorithm requires computation costs and storage capacity of the order $\mathcal{O}(n^2)$, whereas the proposed algorithm requires both computation as well as storage of the order $\mathcal{O}(n)$ Mokhtari & Ribeiro (2015).

### 3.1 CONVERGENCE ANALYSIS

In this section, convergence analysis of the proposed algorithm based on the regret bound approach is performed, and the conditions for a guaranteed convergence based on the bounds on the adaptive learning-rate and the range of the parameter $K$ are determined. Considering $\{J(\theta)\}$ $= \{J(\theta_0), J(\theta_1), J(\theta_2), ..., J(\theta_k)\}$ as the sequence of the loss function values, the regret bound $(R_J(T))$ is the sum of the difference between the optimum value of the loss function, i.e. $J(\theta^*)$ and the series of function values from $J(\theta_0)$ to $J(\theta_k)$, and is given by

$$R_J(T) = \sum_{k=1}^{T} \left[ J(\theta_k) - J(\theta^*) \right] \tag{7}$$

where $J(\theta^*) = \underset{x \in \mathbb{R}}{\arg\min} J(\theta)$. For a convex and twice differentiable loss function $J(\theta) \, \forall \theta \in \mathbb{R}$, whose gradient is $\beta$-Lipschitz continuous i.e., $\forall \, \theta_1, \theta_2 \, \exists \, \| \widehat{\nabla} J\left(\theta_2\right) - \widehat{\nabla} J\left(\theta_1\right) \|_2 < \beta \, \| \theta_2 - \theta_1 \|_2$, the regret bound for the proposed algorithm (proof in Appendix A) is given by

$$R_J(T) \leq \frac{1}{2 \boldsymbol{M}_k} \left[ \, \| \theta_0 - \theta_* \|_2^2 \, \right] \tag{8}$$

where $\boldsymbol{M} = \left[ \mu_k^{(1)}, \mu_k^{(2)}, ..., \mu_k^{(L)} \right]^T$. Further, the algorithm converges with a rate $\mathcal{O}\left(1/\mu_k k\right)$, which means the rate of convergence is inversely dependent on the number of iterations as well as the instantaneous learning-rate $\mu_k$, which in turn is determined from the change in the model-parameter values $\Delta \theta_k$. This implies that, the model parameters on a flatter trajectory will converge at faster rate.

#### 3.1.1 CONVERGENCE BOUNDS AND RANGE OF PARAMETER $K$

In this section, the lower and upper bounds for the adaptive learning-rate in case of a very small or a very large change in model-parameters are determined. The algorithm will converge with a rate $\mathcal{O}(1/k)$, if the adaptive learning-rate is bounded by

$$0 < \mu_k < \frac{2}{\beta} \tag{9}$$

where $k$ the number of iterations and $\beta$ the maximum eigenvalue of the input correlation matrix $\boldsymbol{R}_{\boldsymbol{xx}}$, determined from the classical theoretical analysis of the SGD convergence[1] Boyd & Vandenberghe (2004).

*Case I:* For the model-parameter converging over a very steep slope, i.e., where the value of $\Delta\theta_k^{(i)}$ is very large $(\Delta\theta_k^{(i)} \to \infty)$, the lower bound on the adaptive learning rate is given by

$$\mu_{k\min}^{(i)} = \lim_{\Delta\theta \to \infty} \left[ \mu_0 \cdot \frac{K}{\left(1 + abs\left(\Delta\theta_k^{(i)}\right)\right)} \right] = 0 \tag{10}$$

The minimum adaptive learning-rate $\mu_{k\min}^{(i)}$ for a convergence with a rate $\mathcal{O}(1/k)$, must be bound by $\mu_{k\min}^{(i)} > 0$. Thus, the lower bound of the parameter $K$ is given by $K > 0$.

---

[1] The maximum eigenvalue of the matrix $\boldsymbol{R}_{\boldsymbol{xx}}$ is determined from the trace of the matrix $E\left[\boldsymbol{x}^T\boldsymbol{x}\right]$ and under practical considerations, the trace $\left(E\left[\boldsymbol{x}^T\boldsymbol{x}\right]\right)$ is computed from the average power of the input signal $\boldsymbol{x}$ Haykin (2014).

*Case II:* For the model-parameter following a very slow convergence over a long, gradual slope $(\Delta\theta_k^{(i)} \approx 0)$, the theoretical maximum of the learning-rate is given by

$$\mu_{k\max}^{(i)} = \lim_{\Delta\theta\to 0} \left[ \mu_0 \cdot \frac{K}{\left(1 + abs\left(\Delta\theta_k^{(i)}\right)\right)} \right] = K \cdot \mu_0 \tag{11}$$

and for the gradient descent to have a global convergence, the upper bound must be limited to $\mu_{k\max}^{(i)} \leq 2/\beta$. Thus, the range of the parameters $K$, computed by substituting the minimum and the maximum values respectively, is given by $K \in \left(0, \frac{2}{\mu_0\beta}\right)$.

In this section, the proposed algorithm was described and the bounds on the adaptive learning-rate for a guaranteed convergence were determined. The proposed algorithm provides a novel approach to improve the convergence speed of the gradient descent algorithm, simultaneously guaranteeing a stable convergence.

## 4 EXPERIMENTS

In this section, an extensive experimental analysis was carried out to evaluate the efficacy and performance of the proposed algorithm. The performance of the proposed algorithm was compared with a number of state of the art optimizers like the SGDM, Adam, AdaBelief, LookAhead, RAdam, and AdamW. The first part of the experiments involved implementing the proposed algorithm on a standard two-parameter loss-function, i.e., Rosenbrock function, and visualise and compare its convergence with SGDM and Adam. In the next part, further experiments were carried out implementing the above-mentioned optimizer for training of two benchmark datasets, and comparing their performance based on the training loss and accuracy. The experiments involved training of (i) CIFAR 10 and CIFAR 100 datasets on the ResNet18 architecture, and (ii) Tiny-ImageNet dataset on the EfficientNet architecture.

### 4.1 PERFORMANCE ASSESSMENT ON ROSENBROCK FUNCTION

For the experiments in this section, the above-mentioned algorithms were implemented on a two-dimensional optimization function and their convergence characteristics were compared. The Rosenbrock function is known to be used as a performance evaluation benchmark for different optimization algorithms Emiola & Adem (2021). The Rosenbrock function for a two-parameter setup implemented in the experiments is given by

$$f(w_1, w_2) = \kappa \left(w_1^2 - w_2\right)^2 + (w_1 - 1)^2 \tag{12}$$

where $w_1, w_2 \in \mathbb{R}$ and $f(w_1, w_2)$ is strictly convex and differentiable, and the constant $\kappa$ determines the steepness of the valley of the Rosenbrock function. The minimum of the function lies at $(1, 1)$, and is inside a long, parabolic shaped flat valley, and achieving fast convergence to the minimum becomes difficult in some cases. The Fig. 1(a) shows the Rosenbrock function with a $\kappa = 100$.

The experiment was set up as follows: The model-parameters $w_1$ and $w_2$ were initialized to $(0, 5)$ and the initial learning-rate for all the algorithms was set to $\mu_0 = 0.001$. The hyperparameters $\beta_1$ and $\beta_2$ for the Adam optimizer were set to $\beta_1 = 0.9$ and $\beta_2 = 0.999$ respectively. The performance of the above-mentioned algorithms was analyzed and compared based on the speed of convergence, the accuracy and the run time of the algorithm.

The Figure 1 shows the convergence comparison of the proposed algorithm with the SGD and the Adam optimizers. The plot 1(a) and 1(b) show the 3D surface plot and the contour plot of the convergence of the above-mentioned algorithms, while the plot 1(c) shows the convergence of the function against the number of iterations. It can be seen from the plot 1(c) that the proposed algorithm converges to the minimum in about 25 iterations. In comparison, the SGD algorithm converged in about 40 iterations, while the Adam algorithm showed the slowest convergence for this experiment and required about 160 iterations to converge to minimum. It can be observed that the proposed algorithm outperforms the SGD and the Adam algorithms in terms of the convergence speed, i.e., number of iterations for convergence.

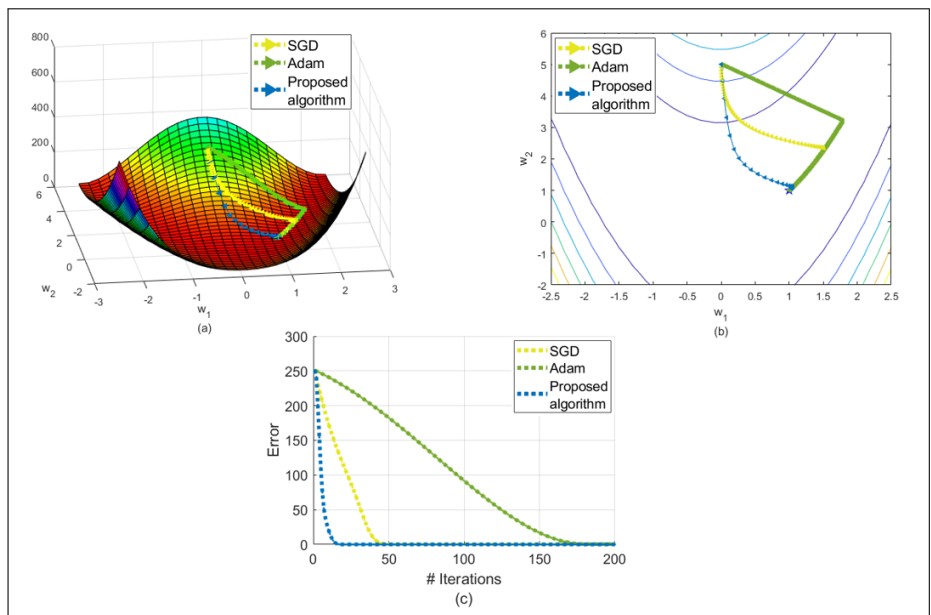

Figure 1: (a) 3D Surface plot of the convergence. (b) 2D contour plot of the convergence. (c) Convergence of the function w.r.t. number of iterations.

## 4.2 PERFORMANCE ASSESSMENT ON NEURAL NETWORKS

### 4.2.1 CIFAR-10 AND CIFAR-100

For the experiments in this section, image classification on the standard CIFAR-10 and CIFAR-100 datasets was performed using the ResNet-18 architecture He et al. (2015). The CIFAR-10 and CIFAR-100 datasets consist of $32 \times 32$ colour images belonging to 10 and 100 classes respectively. The dataset is split into 50000 images for training and 10000 images for test Krizhevsky et al. (2009). The SGDM optimizer was implemented with a step-decay scheduler with an initial learning-rate of 0.1 and decreased by a factor of 10 after every 20 epochs. For the Adam and AdaBelief optimizers, the initial learning rate was chosen as 0.001 with hyperparameters $\beta_1 = 0.9, \beta_2 = 0.999$. The LookAhead optimizer was implemented with a learning-rate of 0.001 and the hyperparameters $\kappa = 5$ and $\alpha = 0.5$. The RAdam and the AdamW optimizers were applied with a learning-rate of 0.001, hyperparameters $\beta_1 = 0.9, \beta_2 = 0.999$ and a weight decay of 0.0001 in case of AdamW. For the proposed algorithm, the initial learning-rate was set at 0.001 and the parameter $K$ was chosen to be $K = 10$. This value was chosen after extensive simulations with different parameter values (refer section 4.2.3) The CNN models were coded on the Google Colaboratory environment and trained on 25 GB RAM GPU and run for 100 epochs for each optimization algorithm with a batch-size of 128.

Table 1: Comparison of the training loss and accuracy for the CIFAR Datasets

| Optimizer | Dataset | | | |
| --- | --- | --- | --- | --- |
| | CIFAR 10 | | CIFAR 100 | |
| | Loss | Accuracy | Loss | Accuracy |
| SGDM | 0.06139 | 0.98886 | 0.28875 | 0.93812 |
| Adam | 0.00936 | 0.99713 | 0.14823 | 0.96270 |
| AdaBelief | 0.04569 | 0.99080 | 0.26306 | 0.94918 |
| LookAhead | 0.37017 | 0.87695 | 0.01018 | 0.99905 |
| RAdam | 0.01806 | 0.99617 | 0.16338 | 0.96631 |
| AdamW | 0.01783 | 0.99422 | 0.02219 | 0.99320 |
| Proposed algorithm | **0.00012** | **1.0** | **0.00571** | **0.99914** |

The figure 2(a) shows the training results for the classification task on the CIFAR-10 dataset, and the figure 2(b) for the CIFAR-100 dataset. It can be observed that for all algorithms can achieve an accuracy approaching almost 99%. The proposed algorithm performs slightly better in case of both the CIFAR-10 and the CIFAR-100 datasets. The performance evaluation based on the training accuracy and loss for each optimizer is compiled in the table 1.

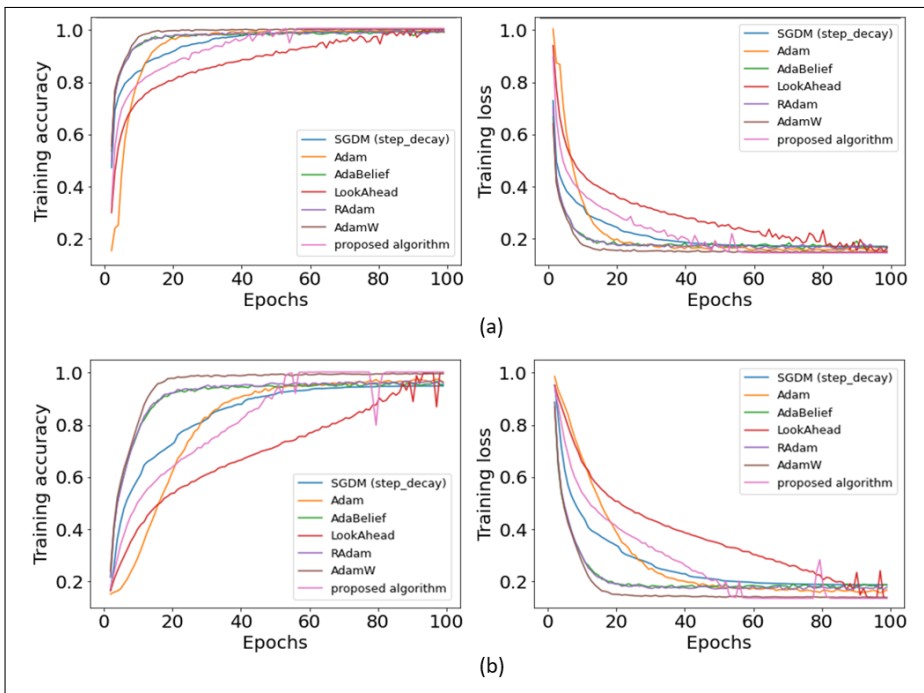

Figure 2: (a) Convergence comparison on the CIFAR-10 dataset (b) Convergence comparison on the CIFAR-100 dataset

### 4.2.2 TINY-IMAGENET

For the next experiments, the Tiny-ImageNet dataset was implemented Le & Yang (2015) and the training was carried out using the EfficientNetB3 architecture Tan & Le (2019). This dataset consists of 100,000 training images, 10,000 validation images, and 10,000 testing images and are divided into 200 classes. The images are extracted from the ImageNet dataset, cropped and resized to be $64 \times 64$. Due to the heavy computational burden, the simulation comparisons were carried out only with SGDM and the Adam algorithm. The hyperparameters for the SGDM (with step-decay), Adam optimizer and the proposed algorithm were the same as in the above experiments. Figure 3 shows the learning curves for each optimization method for both the training accuracy and training loss on the Tiny-ImageNet dataset. It can be observed that both the Adam and the proposed algorithm perform better than the SGDM and the proposed algorithm shows slightly better performance than the Adam algorithm. The Table 2 summarizes the training and test accuracy and loss for the above mentioned model in case of each optimizer.

### 4.2.3 EXPERIMENTS WITH DIFFERENT HYPERPARAMETERS

In this section, the performance of the proposed algorithm was evaluated for different hyperparameters applied on the CIFAR-10 dataset. The proposed algorithm was implemented for training the dataset for different values of the parameter $K$, i.e., 0.1, 1, 5, 10, and 20. It can be observed that for smaller values of the parameter $K$, the algorithms shows slower convergence and results in lower accuracy. However, convergence of the algorithm is aggressive for larger values of $K$ and for $K = 20$ shows large oscillations at the middle stage of training. Training with parameter $K = 10$ shows the best overall performance. The figure 4 shows the comparisons of the training curves for the different hyperparameters and the table 3 lists the training accuracy and training loss for the same.

Table 2: Training and Test Loss and accuracy on the Tiny-ImageNet dataset

| Tiny-ImageNet | | |
|---|---|---|
| **Optimizer** | **Loss** | **Accuracy** |
| SGDM | 1.21919 | 0.69322 |
| Adam | 0.71841 | 0.79568 |
| Proposed algorithm | **0.69584** | **0.80373** |

Table 3: Training and Test Loss and accuracy for different values of parameter $K$

| CIFAR-100 | | |
|---|---|---|
| **Parameter** | **Loss** | **Accuracy** |
| K = 0.1 | 0.71115 | 0.75323 |
| K = 1 | 0.21901 | 0.92940 |
| K = 5 | 0.0010 | 1.0 |
| K = 10 | **0.00012** | **1.0** |
| K = 20 | 4.35486 | 1.0 |

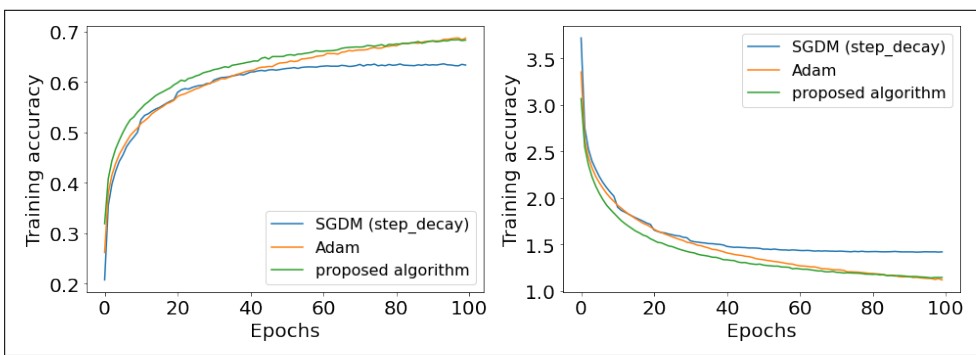

Figure 3: Convergence comparison on the Tiny-ImageNet dataset

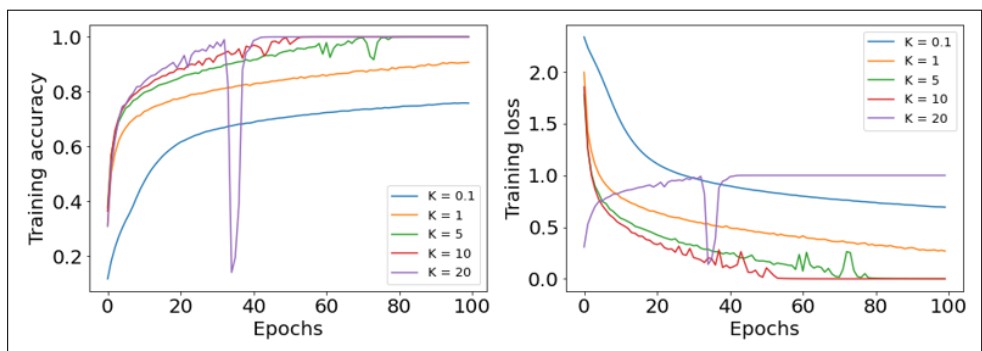

Figure 4: Convergence comparison for different hyperparameters $K$

## 5 CONCLUSION

In this paper, a new approach to gradient descent optimization is presented, which updates the learning-rate of the algorithm for each model-parameter, inversely proportional to the change in model-parameter values from the preceding iteration. The algorithm adapts the model-parameters on gentle slope with larger learning-rate relative to model-parameters on steep slope, thereby improving the overall convergence speed. The efficacy of the proposed algorithm was proven based on the training of standard datasets like the CIFAR-10/100 and the Tiny-ImageNet datasets, and it was noted that the proposed algorithm surpasses the currently popular algorithms. Future work involves deeper analysis of the proposed algorithm on complex non-convex loss functions and implementing it on other benchmark datasets and models.

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

## A    APPENDIX

The proof of convergence of the algorithm is carried out considering the regret bound of the algorithm. Consider $J(\theta)$ as the objective function to be optimized and consider $\{J(\theta)\} = \{J(\theta_0), J(\theta_1), J(\theta_2), ..., J(\theta_k)\}$ as the sequence of the loss function values. Regret bound, given by $R_J(T)$, is the sum of the difference between the optimum value of the loss function, i.e. $J(\theta^*)$ and the series of function values from $J(\theta_0)$ to $J(\theta_k)$. The regret bound is given by

$$R_J(T) = \sum_{k=1}^{T} [J(\theta_k) - J(\theta^*)] \tag{13}$$

where $J(\theta^*) = \underset{x \in \mathbb{R}^C}{\arg\min} J(\theta)$

**Notations:** $J(\theta_k^{(i)})$ gives the value of the loss function at instance $k$ with respect to the model-parameter $i$. $\nabla J(\theta_k^{(i)})$ gives the gradient of the loss function at instance $k$ with respect to the model-parameter $i$.

**Assumptions:** For the convergence analysis, the following assumptions are made:

*Assumption 1:* The function $J(\theta)$ is convex, i.e.,

$$J(x_2) \geq J(x_1) + \langle \nabla J(x_1), (x_2 - x_1) \rangle \quad \forall x_1, x_2 \in \mathbb{R}^n \tag{14}$$

*Assumption 2:* $\nabla J(\theta)$ is $L-$ Lipschitz continuous, i.e.,

$$\| \nabla J(x_2) - \nabla J(x_1) \| < L \| x_2 - x_1 \| \quad \forall x_1, x_2 \in \mathbb{R}^n \tag{15}$$

and

$$J(x_2) \leq J(x_1) + \langle \nabla J(x_1), (x_2 - x_1) \rangle + \frac{L}{2} \| x_2 - x_1 \|_2^2 \quad \forall x_1, x_2 \in \mathbb{R}^n \tag{16}$$

**Proof:** The convergence analysis attempts to prove that the upper bound of the regret function $R_J(T)$ is bounded by the inverse of the number of iterations.

Considering the Lipschitz continuity, we have

$$J(\theta_2) \leq J(\theta_1) + \langle \nabla J(\theta_1), (\theta_2 - \theta_1) \rangle + \frac{L}{2} \| \theta_2 - \theta_1 \|_2^2 \quad \forall \theta_1, \theta_2 \in \mathbb{R}^n \tag{17}$$

for the $i^{th}$ model-parameter, the above equation can be written as

$$J(\theta_{k+1}^{(i)}) \leq J(\theta_k^{(i)}) + \left\langle \nabla J(\theta_k^{(i)}), \left( \theta_{k+1}^{(i)} - \theta_k^{(i)} \right) \right\rangle + \frac{L}{2} \| \theta_{k+1}^{(i)} - \theta_k^{(i)} \|_2^2 \tag{18}$$

$$J(\theta_{k+1}^{(i)}) \leq J(\theta_k^{(i)}) + \left\langle \nabla J(\theta_k^{(i)}), -\mu_k^{(i)} \nabla J(\theta_k^{(i)}) \right\rangle + \frac{L}{2} \| -\mu_k^{(i)} \nabla J(\theta_k^{(i)}) \|_2^2 \tag{19}$$

$$\leq J(\theta_k^{(i)}) + \left\langle \nabla J(\theta_k^{(i)}), -\mu_k^{(i)} \nabla J(\theta_k^{(i)}) \right\rangle + \frac{L}{2} \| -\mu_k^{(i)} \nabla J(\theta_k^{(i)}) \|_2^2 \tag{20}$$

$$\leq J(\theta_k^{(i)}) - \mu_k^{(i)} \| \nabla J(\theta_k^{(i)}) \|_2^2 + \frac{(\mu_k^{(i)})^2 L}{2} \| \nabla J(\theta_k^{(i)}) \|_2^2 \tag{21}$$

$$\leq J(\theta_k^{(i)}) - \mu_k^{(i)} \left( 1 - \frac{\mu_k^{(i)} L}{2} \right) \| \nabla J(\theta_k^{(i)}) \|_2^2 \tag{22}$$

using the condition $\mu_k^{(i)} \leq \frac{1}{L}$ and plugging it in 22

$$J(\theta_{k+1}^{(i)}) \leq J(\theta_k^{(i)}) - \frac{\mu_k^{(i)}}{2} \parallel \nabla J(\theta_k^{(i)}) \parallel_2^2 \tag{23}$$

It can be observed from the above Eq. 23 that since $\parallel \nabla J(\theta_k^{(i)}) \parallel_2^2$ is always positive, the value of the loss function $J(\theta_k^{(i)})$ always decreases after every iteration untill it reaches the minimum $J(\theta_*^{(i)})$, where the gradient of the loss function $\nabla J(\theta^{(i)}) = 0$.

Now considering the convexity of the loss function $\nabla J(\theta)$ again (*Assumption 1*), the following can be written

$$J(\theta_*^{(i)}) \geq J(\theta_k^{(i)}) + \left\langle \nabla J(\theta_k^{(i)}), \left( \theta_*^{(i)} - \theta_k^{(i)} \right) \right\rangle \tag{24}$$

$$\Rightarrow J(\theta_k^{(i)}) \leq J(\theta_*^{(i)}) + \left\langle \nabla J(\theta_k^{(i)}), \left( \theta_k^{(i)} - \theta_*^{(i)} \right) \right\rangle \tag{25}$$

Plugging the above in Eq. 23 gives

$$J(\theta_{k+1}^{(i)}) \leq J(\theta_*^{(i)}) + \nabla J(\theta_k^{(i)}) \parallel \theta_k^{(i)} - \theta_*^{(i)} \parallel_2 - \frac{\mu_k^{(i)}}{2} \parallel \nabla J(\theta_k^{(i)}) \parallel_2^2 \tag{26}$$

$$J(\theta_{k+1}^{(i)}) \leq J(\theta_*^{(i)}) + \nabla J(\theta_k^{(i)}) \parallel \theta_k^{(i)} - \theta_*^{(i)} \parallel_2 - \frac{\mu_k^{(i)}}{2} \parallel \nabla J(\theta_k^{(i)}) \parallel_2^2$$
$$+ \frac{1}{2\mu_k^{(i)}} \left( \parallel \theta_k^{(i)} - \theta_*^{(i)} \parallel_2^2 - \parallel \theta_k^{(i)} - \theta_*^{(i)} \parallel_2^2 \right) \tag{27}$$

Rearranging the above equation gives

$$J(\theta_{k+1}^{(i)}) - J(\theta_*^{(i)}) \leq \frac{1}{2\mu_k^{(i)}} \Big[ 2\mu_k^{(i)} \nabla J(\theta_k^{(i)}) \parallel \theta_k^{(i)} - \theta_*^{(i)} \parallel_2 (\mu_k^{(i)})^2 \parallel \nabla J(\theta_k^{(i)}) \parallel_2^2$$
$$- \parallel \theta_k^{(i)} - \theta_*^{(i)} \parallel_2^2 + \parallel \theta_k^{(i)} - \theta_*^{(i)} \parallel_2^2 \Big] \tag{28}$$

which on simplification gives,

$$J(\theta_{k+1}^{(i)}) - J(\theta_*^{(i)}) \leq \frac{1}{2\mu_k^{(i)}} \Big[ \parallel \theta_k^{(i)} - \theta_*^{(i)} \parallel_2^2 - \left( \theta_k^{(i)} - \theta_*^{(i)} - \mu_k^{(i)} \nabla J(\theta_k^{(i)}) \right) \Big] \tag{29}$$

$$\Rightarrow J(\theta_{k+1}^{(i)}) - J(\theta_*^{(i)}) \leq \frac{1}{2\mu_k^{(i)}} \Big[ \parallel \theta_k^{(i)} - \theta_*^{(i)} \parallel_2^2 - \parallel \theta_{k+1}^{(i)} - \theta_*^{(i)} \parallel_2^2 \Big] \tag{30}$$

The above inequality hold for every iteration of the sequence. Thus, summing over all the iterations, gives

$$\sum_{i=0}^{k-1} J(\theta_{k+1}^{(i)}) - J(\theta_*^{(i)}) \leq \frac{1}{2\mu_k^{(i)}} \sum_{i=0}^{k-1} \Big[ \parallel \theta_k^{(i)} - \theta_*^{(i)} \parallel_2^2 - \parallel \theta_{k+1}^{(i)} - \theta_*^{(i)} \parallel_2^2 \Big] \tag{31}$$

The sum element on the right-hand side of the equation is a telescopic sum, and hence all the intermediate terms disappear, resulting in the following

$$\sum_{i=0}^{k-1} J(\theta_{k+1}^{(i)}) - J(\theta_*^{(i)}) \leq \frac{1}{2\mu_k^{(i)}} \left[ \parallel \theta_0^{(i)} - \theta_*^{(i)} \parallel_2^2 - \parallel \theta_k^{(i)} - \theta_*^{(i)} \parallel_2^2 \right] \tag{32}$$

The second term in the right-hand side, i.e., $\parallel \theta_k^{(i)} - \theta_*^{(i)} \parallel_2^2$ is always positive and hence the following inequality holds true as well

$$\sum_{i=0}^{k-1} J(\theta_{k+1}^{(i)}) - J(\theta_*^{(i)}) \leq \frac{1}{2\mu_k^{(i)}} \left[ \parallel \theta_0^{(i)} - \theta_*^{(i)} \parallel_2^2 \right] \tag{33}$$

This relation shows that for a step-size $\mu_k^{(i)} \leq \frac{1}{L}$, the regret of the bound of the gradient descent algorithm is bounded by

$$R_J^{(i)}(T) \leq \frac{1}{2\mu_k^{(i)}} \left[ \parallel \theta_0^{(i)} - \theta_*^{(i)} \parallel_2^2 \right] \tag{34}$$

It was established in the above section that the upper bound of the variable learning-rate at any instance is bounded by $\mu_k^{(i)} \in (0, \mu_0)$ and it can be inferred that the regret bound of the algorithm is given by

$$R_J^{(i)}(T) \leq \frac{1}{2\mu_k^{(i)}} \left[ \parallel \theta_0^{(i)} - \theta_*^{(i)} \parallel_2^2 \right] \leq \frac{1}{2\mu_0} \left[ \parallel \theta_0^{(i)} - \theta_*^{(i)} \parallel_2^2 \right] \tag{35}$$

And the regret bound of the complete loss function can be computed by aggregating the above relation across all dimensions, for all the model-parameters.

$$R_J(T) \leq \frac{1}{2\boldsymbol{M}_k} \left[ \parallel \theta_0 - \theta_* \parallel_2^2 \right] \tag{36}$$

where $\boldsymbol{M} = \left[ \mu_k^{(1)}, \mu_k^{(2)}, ..., \mu_k^{(L)} \right]^T$.

where $\boldsymbol{M} = \left[ \mu_k^{(1)}, \mu_k^{(2)}, ..., \mu_k^{(L)} \right]^T$. Further, the algorithm converges with a rate $\mathcal{O}\left(1/\mu_k k\right)$, which means the rate of convergence is inversely dependent on the number of iterations as well as the instantaneous learning-rate $\mu_k$, which in turn is determined from the change in the model-parameter values $\Delta\theta_k$. This implies that, nearing convergence, when the change in model-parameter values is very small, the algorithm converges faster than vanilla-GD (or vanilla-SGD).

Since it is assumed that the loss function $J(\theta)$ is convex and the function $J(\theta)$ is reducing at every iteration for each model-parameter, i.e., $J(\theta_{k+1}) < J(\theta_k) < J(\theta_{k-1}) < ....$

$$J(\theta_{k+1}) \leq \frac{1}{k} \sum_{i=0}^{k-1} [J(\theta_k)]$$

$$\Rightarrow J(\theta_{k+1}) - J(\theta_*) \leq \frac{1}{k} \sum_{i=0}^{k-1} [J(\theta_k) - J(\theta_*)]$$

Thus, it can be concluded that the algorithm converges with a rate $O\left(\frac{1}{k}\right)$, given by

$$\Rightarrow J(\theta_{k+1}) - J(\theta_*) \leq \frac{1}{k} \frac{1}{2\boldsymbol{M}_k} \left[ \parallel \theta_0 - \theta_* \parallel_2^2 \right] \tag{37}$$

