# OpenReview forum: "Improved Gradient Descent Optimization Algorithm based on Inverse Model-Parameter Difference"
_ICLR.cc/2023/Conference — Submitted to ICLR 2023_

### Official Review · Reviewer_xzMY · 2022-10-13

**Confidence:** 5
**Correctness:** 1
**Technical Novelty And Significance:** 2
**Empirical Novelty And Significance:** Not applicable
**Recommendation:** 1

**Clarity, Quality, Novelty And Reproducibility:**

### Clarity
The paper is written clearly and the algorithm is explained in full detail. However, some of the reasons used to motivate the method seem unsatisfying to me.

### Quality
The mathematical sections of the paper are simply wrong. The experimental comparison has several shortcomings.

### Originality
The method is, to the best of my knowledge, novel. Of course, papers on new variants of adaptive gradient algorithms for deep learning operate in a narrow space.

**Strength And Weaknesses:**

### Strenghts

1) The paper follows a clear structure and is generally well-written.

2) I like the illustration of the proposed method on the Rosenbrock function.

### Weaknesses

1) Generally, I find the motivation offered for the method unsatisfying. Some of the reasons used to motivate the method are not really substantiated. For example, on page 2, the paper states that the proposed algorithm avoids
"the suboptimal convergence due to very large or very small scaling of the learning rate."
While this might have been some intuition guiding the authors, the do not offer any reason as to why very large or very small scaling is inherently bad. In fact, the literature on
adaptive gradient methods probably argues that drastic step size differences in different coordinates are a _feature_ not a bug.

2) The parametrization of the proposed algorithm is redundant. In Eq. (4), the hyperparameter
$K$ could simply be subsumed into the "global" learning rate $\mu_0$. (This also has some implications for the experimental comparison, see point 6.)
On the other hand,
one could imagine having an additional hyperparameter in the denominator.
If I were to experiment with an update of the proposed form, I would want to parametrize
it as $a / (1 + b \Delta)$.

3) Neither the regret bound in in Eq. (8)
nor its proof (Appendix A) make mathematical sense. The cumulative regret on the left-hand
side of Eq. (8) is a scalar quantity. The right-hand side seems to be $k$-dimensional vector.
In the proof, Eq. (18) does not have any well-defined mathematical meaning.
$J$ is a function of _all_ model parameters
$\theta\in \mathbb{R}^d$, so it is mathematically meaningless to plug in only the $i$-th model
parameter $\theta^{(i)}$.
It seems the authors want to assume some sort of decomposition across individual coordinates,
but this would certainly require additional assumptions beyond smoothness and convexity.
Even assuming that some such decomposition exists, there are a more errors in the
remainder of the proof.
I'm not going to list all of them, but for example going from Eq. (25)-(26) is not
possible and using Cauchy-Schwarz would introduce a norm around the gradient term.

4) In Section 3.1.1, the paper states that the algorithm will converge if $\mu_k< 1 / (2 \beta)$
where $\beta$ is "the maximum eigenvalue of the input correlation matrix".
This statement is not applicable to the setting the paper is assuming (smoothness + convexity). It might hold for
linear least-squares regression, but if that is what the authors are referring to here,
it should be stated very clearly.

5) In the beginning of of Section 4.1, it is falsely stated that the Rosenbrock function is strictly convex.

6) The experimental comparison has several shortcomings in my opinion. The base learning rate is
set to the same value for _all_ methods. For the proposed method, the hyperparameter $K$
is tuned (the tuning protocol employed is unclear). Since $K$ is simply a multiplier for the
learning rate (see comment 2) this amounts
to an unfair comparison.
Furthermore, the experiments seem to have been conducted with a single random seed.
Multiple seeds should be used to gauge the variabilty of the performance.
Finally, the paper compares _training_ loss and accuracy. While this is adequate from an
optimization perspective, the potential users of the method will be more interested in the
performance on the test set.

**Summary Of The Paper:**

The paper proposes an adaptive stochastic gradient method, which scales element-wise step sizes inversely proportional to the update magnitude in the previous step. A regret bound is provided and the method is evaluated on a number of neural network training tasks.

**Summary Of The Review:**

In light of the weaknesses discussed above, I recommend rejecting this paper. The mathematical sections contain multiple severe errors (weakness 3-4) and the experimental comparison is not up to the standards that are expected of empirical work on optimization methods (weakness 6). I would also question the basic motivation for the proposed methods (weakness 1), but of course it is perfectly justifiable to just "try out" something.

To the authors, I want to say that my negative review is not meant to diminish your work or to discourage you from pursuing it further. The mathematical errors might be fixable and the experimental comparison can be improved (clear tuning protocol, multiple random seeds, report test performance).

---

### Official Review · Reviewer_Pa1W · 2022-10-25

**Confidence:** 4
**Correctness:** 2
**Technical Novelty And Significance:** 2
**Empirical Novelty And Significance:** 2
**Recommendation:** 3

**Clarity, Quality, Novelty And Reproducibility:**

Clarity:
The paper is overall easy to follow.

Novelty:
Marginal

Originality:
Marginal

**Strength And Weaknesses:**

Strength:
The paper is overall easy to follow and well presented. The authors conducted analysis in both theory and validation on toy and real-world datasets.

Weakness:
1. The proposed method is actually counter-intuitive. If $abs(\Delta \theta)$ is large in dimension $I$, for SGD it means the gradient amplitude in dimension $i$ is large. I don't see why the update stepsize should be small in the $i-th$ dimension.
2. The proof of convergence is restricted to convex objective functions, instead of the more general stochastic non-convex optimization setting. Therefore the results in pretty limited.
3. Experimental validation on Cifar datasets is far from satisfactory. The authors should validate the proposed method in various tasks with different model architectures and datasets.

**Summary Of The Paper:**

The authors proposed a new optimizer, which adapts the stepsize by $K/(1+abs(\Delta \theta))$ where $\Delta \theta$ is the update of the parameters. The authors proved the convergence of the proposed algorithm and conducted experimental validations.

**Summary Of The Review:**

The proposed method is counter-intuitive, and the proof of convergence is only limited to deterministic convex objective functions, and the experimental validations are pretty naive.

---

### Official Review · Reviewer_D864 · 2022-10-25

**Confidence:** 2
**Clarity, Quality, Novelty And Reproducibility:** N/A
**Correctness:** 2
**Technical Novelty And Significance:** 2
**Empirical Novelty And Significance:** 1
**Recommendation:** 1

**Strength And Weaknesses:**

Pros:
The idea is easy to understand.

Cons:
1. The notation is not presented very clearly. For example, the explaination of $K$ in eqn.(3) appears very lately.
2. The idea is a little strange to me. In my understanding, the adaptive learning methods (including Nestrov accelerated momentum) should aim for estimate the Hessian to obtain an approximation of the best step-size in comparision with second-order methods. For example, the momentum term of NAG could be reformulated to the difference of gradients between two recent iterations, where is a reasonable approximation for the second-order information. However, it makes no sense to me that the difference of model parameters could do the similar thing, therefore I doubt the theoritical correctness of the proposed method.
3. It's quite strange that only evaluating the training accuracy but not validation / test accuracy on real datasets, especially for stochastic optimization methods. If we only aim for training, then gradient decent should be the best method. Also, the training stability of the proposed method seems very badly, therefore I doubt the practical usage of the proposed method.

**Summary Of The Paper:**

This paper proposes a new adaptive learning rate first-order optimizer, where the learning rate is scaled by the inverse parameter difference between two recent iterations. Theoretical analysis based on regret bound is presented, and experimental results on image dataset / CNN models are conducted.

**Summary Of The Review:**

I have low confidence of this review, and happy to take advices from other reviewers, and AC.

Update: after carefully reading other reviews, I generally agree reviewer xzMY's comment, and appreciate his/her efforts on checking the correctness of proof. I lower down my score to clear reject.

---

### Official Review · Reviewer_LcE9 · 2022-10-31

**Confidence:** 4
**Correctness:** 2
**Technical Novelty And Significance:** 2
**Empirical Novelty And Significance:** 2
**Recommendation:** 3

**Clarity, Quality, Novelty And Reproducibility:**

The clarity and quality of the paper is low, as the notation system is messy.

Novelty is limited as the idea of adaptive learning is widely explored.

The idea is easy to implement, so it should be reproducible.

**Strength And Weaknesses:**

Strength:

1. The idea is simple and easy to understand. It is also easy to implement.

Weaknesses:

1. There are a bunch of notation errors/overloading in the paper that make it hard to follow. Examples:

    1.1. Vectors should be in $\mathbb{R}^L$ rather than $\mathbb{R}$, e.g.,  $p, q \in \mathbb{R}$, $\theta \in \mathbb{R}$, etc.

    1.2. The equation immediately after eqn (7) should be $J(\theta^*)= \arg\min_\theta J(\theta)$.

    1.3. While eqn (8) uses $M_k$, the following sentence uses $M$. And there is no definition about $\mu_k$ in the next sentence.

    1.4. \beta is used for the Lipschitz constant above eqn (8), and then overloaded as the maximum eigenvalue of the input correlation matrix in eqn(9).

2. Given the the notation systems are so messy, it is hard to believe the proof is correct.

3. The algorithm involves parameters $K$ and $\mu_0$, but there is no guidance on how to choose them.

4. The empirical studies are not convincing either.

    4.1. In Figure 2, the proposed algorithm converges obviously slower than quite a few other algorithms. Although it seems to converge to a point with lower loss, it is unclear why it is the case. Also, the proposed algorithm is unstable compared to others, as its curve is bumpy.

    4.2. The algorithm is only evaluated on three small image datasets. It is unclear if the conclusion can generalize to other domains/networks.

**Summary Of The Paper:**


This paper proposes an adaptive learning rate method to improved the stochastic gradient descent algorithm by leveraging the difference of two consecutive iterations. More concretely, the authors proposes that the learning rate of each parameter should be inversely proportional to the difference between the current and the previous iteration of the this parameter. The authors claim the such an updating scheme, when combined with SGD, can converge under mild assumptions. Empirical studies also show that it can achieve lower training loss and better accuracy on CIFAR-10/100 and Tiny-ImageNet.

**Summary Of The Review:**

The paper is not well written that it is hard to tell if the proof is correct. Besides, the idea is not very novel and the empirical study is not comprehensive either, making it less convincing that the proposed algorithm is effective.

---

### Decision · Program_Chairs · 2023-01-20

**Decision:**

Reject

**Justification For Why Not Higher Score:**

No reviewer voted for accept

**Justification For Why Not Lower Score:**

N/A

**Metareview: Summary, Strengths And Weaknesses:**

Dear authors,

There was unanimous agreement between the reviewers that the paper is not ready for publication as there are multiple errors or misunderstandings.